

# Odontometric analysis of permanent mandibular first and second premolars in an Indian population using cone beam computed tomography

Fiza Hamdulay[1], Ajinkya M. Pawar[1], Shivani Singh[2], Suraj Arora[3], Luca Testarelli[4], Asok Mathew[5,6], Alexander Maniangat Luke[5,6] and Mohmed Isaqali Karobari[7]

[1] Department of Conservative Dentistry and Endodontics, Nair Hospital Dental College, Mumbai, Maharashtra, India
[2] Department of Oral Medicine and Radiology, Nair Hospital Dental College, Mumbai, Maharashtra, India
[3] Department of Restorative Dental Sciences, College of Dentistry, King Khalid University, Abha, Saudi Arabia
[4] Department of Oral and Maxillo-Facial Sciences, Sapienza University of Rome, Rome, Italy
[5] Department of Clinical Science, College of Dentistry, Ajman University, Al-Jruf, Ajman, United Arab Emirates
[6] Centre of Medical and Bio-Allied Health Science Research, Ajman University, Al-Jruf, Ajman, United Arab Emirates
[7] Department of Conservative Dentistry and Endodontics, Saveetha Dental College and Hospital, Saveetha Institute of Medical and Technical Sciences, Chennai, Tamil Nadu, India

Corresponding author
Ajinkya M. Pawar,
ajinkya@drpawars.com

## ABSTRACT

**Background**. Differences in tooth size across populations can significantly influence dental diagnosis, treatment planning, and forensic identification. Despite its relevance, comprehensive region-specific odontometric data—particularly for the Indian population—remain limited. This study aimed to obtain odontometric measurements of permanent mandibular first and second premolars in the Indian population using cone beam computed tomography (CBCT), and to compare these measurements with Wheeler's dental morphology standards, focusing on crown length, root length, and total tooth length.

**Methods**. In this retrospective study, 300 high-quality CBCT scans were analyzed. Measurements of crown, root, and overall tooth lengths for mandibular first and second premolars were obtained using CS 3D Imaging (Version 3.9.3). A one-sample $t$-test was performed to compare the mean values with Wheeler's reference standards.

**Results**. Statistically significant differences were observed in all measured dimensions ($p < 0.001$). For first premolars, crown, root, and total lengths were shorter by 2.3 mm, 1.1 mm, and 3.4 mm, respectively. Second premolars showed reductions of 1.9 mm, 1.2 mm, and 3.0 mm, respectively.

**Conclusions**. This odontometric analysis highlights distinct dental morphological characteristics in the Indian population. These findings offer valuable insights for applications in anthropology, evolutionary biology, forensic science, and clinical dentistry.

## INTRODUCTION

Dentistry incorporates odontometry as a field of inquiry in measurement and analysis of dental characteristics such as crown or root measurements. Odontometry, as stated, serves as a source of information for treatment planning in orthodontics, prosthodontics, and endodontics, and contributes to forensic identification and anthropological studies (*Krishan, Kanchan & Garg, 2015*; *Pillai, 2018*). Knowledge of dental morphology (Odontometry) also helps identify deviations in morphology that will influence a clinical procedure (*Sivakumar, Thangaswamy & Ravi, 2012*; *Calheiros-Lobo, Costa & Pinho, 2022*). In an educational enterprise, odontometric data can aid in education in drawing and carving teeth, and for training students to prepare modified tooth structure for endodontic therapy procedures—the student must obtain measurement accuracy using odontometric techniques (*Giuliani et al., 2007*; *Alovisi et al., 2020*; *Overskott et al., 2024*).

Odontometry is a vital tool in forensic sciences and has unique advantages because teeth are one of the most durable biological materials after death. In light of the high durability of teeth when composing an identification reference in mass fatalities or legitimate investigations, *Issrani et al. (2022)* explored major forensic implications in odontometry. Additionally, in anthropology, the systematic measurements of dental morphology are valuable studies that not only measure the study of traits or ethnic variations but also evolutionary patterns of human beings, as noted by *Jayakrishnan, Reddy & Vinod Kumar, (2021)*. These vital studies cannot be understated in their contributions to the forensic sciences and anthropology.

Traditional odontometry methods in the past had certain factors undermine reliability in regard to reproducibility and distortion. Cone beam computed tomography (CBCT) presented an overall better method, as it provided high resolution three dimensional (3D) reconstructions that eliminated concerns of reproducibility and distortion (*Venkatesh & Elluru, 2017*). CBCT offered a different view of the anatomy allowing accurate, undistorted views of crown and root anatomy which could benefit both clinical and research applications (*Christiaens et al., 2023*). Recently, CBCT applications have expanded into forensic odontology, allowing for substantially improved accuracy in the postmortem identification and creation of digital dental records (*Corte-Real et al., 2024*). In anthropology, CBCT based data were used to measure morphological characteristics of members of different populations, resulting in genetic diversity and evolutionary adaptation applications (*Ajmal et al., 2023*; *Popowics & Mulimani, 2023*).

It is essential to appreciate the unexploited abilities of CBCT for population-specific odontometric data, despite its relevance to the Indian population. There is almost a cling toward reference standards universally accepted such as Wheeler's, but Wheeler's standard represents Western people and does nothing to represent Indian people and the diverse anatomical and anthropological features that occur in the Indian population. Making reference odontometric values specific to the native populations is not only important but also necessary if we want any great precision for clinical, forensic, and anthropological reasons; and need to find tacit acceptance of accuracy in our scientists', practitioners', and healthcare professionals' practices and behaviours.

The first and second mandibular premolars are certainly important in clinical dentistry due to their essential roles in mastication, occlusal balance and orthodontic anchorage (*Albuquerque, Kottoor & Hammo, 2014*). The variation in the morphology of these teeth is more than an aesthetic detail and actually reflects a critical role affecting the success of restorative design and treatment accuracy (*Sierpinska et al., 2017*). An incorrect measurement in crown or root length and/or misidentifying any of the premolar features is not simply an error (*Chate, 2012*; *Li, Parada & Chai, 2017*; *Nagaş, Eğilmez & Kivanç, 2018*), but rather a significant error that can impact endodontic success and compromise orthodontic services. It is essential that dental practitioners account for accurate measures of a patient's first and second premolars to achieve the best patient care and treatment success.

With this background, this study aimed to enhance and potentially redefine our understanding of odontometric parameters (*i.e.,* crown length, root length and total length) of mandibular first and second premolars in the Indian population *via* the enhanced application of CBCT imaging. The comparisons across these measurements will take into account Wheeler's standard and widely accepted reference data and will help ascertain statistically significant values to determine the validity of the null hypothesis. This study hopes to provide current, population-specific norms for the profession to improve accuracy in clinical practices, elevate dental education, advance the field of forensic odontology, and provide grounds for advancing anthropological sciences. This research will not only address an important gap in region-specific dental profession data, but also lay the groundwork for a transition to more accurate evidence-based practices in many areas.

## MATERIALS & METHODS

The Institutional Ethics Committee at Nair Hospital Dental College granted approval for the retrospective investigation, which was executed in accordance with the Declaration of Helsinki (Clearance ID: EC-208/CONS/ND111/2023; Approval Date: 01-03-2023). The Ethics Committee also waived the need for participants' informed permission considering the study was retrospective in nature and used anonymized data, protecting participants' privacy and confidentiality while upholding ethical standards.

### Sample size estimation and acquisition

The sample size was determined by considering an expected occurrence rate of the outcome factor in the population, estimated at 27.8%, with a confidence interval of $\pm 5\%$. Assuming a large population size ($N = 1,000,000$) and a design effect (DEFF) of 1 to account for simple random sampling, a sample size of 300 was calculated to achieve a 95% confidence level. The calculation was performed using the formula: n = [DEFF*Np(1−p)]/[(d2/Z 21−$\alpha$/2*(N−1)+p*(1−p)]. This approach was based on the methodology employed in a previous study (*Kulkarni et al., 2020*), which utilized a similar method for population-based anatomical assessments using cone-beam computed tomography.

The patients who attended the Department of Oral Medicine, Diagnosis, and Radiology at Nair Hospital Dental College in Mumbai between March 2023 and October 2023 furnished the study sample for the CBCT scans. CBCT images were obtained as part of

routine radiographic assessments for a range of dental treatments, utilizing a NewTom VGi scanner (Verona, Italy). CBCT scans were chosen at random from adult patients over the age of 18, including 172 males and 128 females, who had undergone standard dental imaging procedures. There was no stratification based on gender, dental health, or socioeconomic background. Only those scans that fulfilled the inclusion criteria outlined below were included. Ttechnical parameters used were: 110 kV, 0.3–2 mA, range mAs 2.5–6.7, scan time <12 s, FOV of 12 × 8 cm or 12 × 15 cm. Voxel size was 0.25 mm and slice thickness of axial images was 0.25 mm. The delivered dose was 2.0–2.2 mGy ± 30%. The images were created in DICOM format and evaluated by axial, cross- sectional and sagittal reconstructions with a thickness of one mm and a cutting interval of one mm.

Achieving exceptional image quality in CBCT scans was of utmost importance, and our comprehensive evaluation process ensured this high standard. Each scan underwent a detailed assessment based on strict quality control criteria, which required (1) the complete elimination of motion artifacts, (2) the clear visibility of essential anatomical features like the cementoenamel junction (CEJ) and root apex, and (3) a uniform voxel size free from any distortion. This thorough evaluation was carried out by two experienced examiners (A.M.P. and S.S.), whose expertise guaranteed the highest level of scrutiny. Scans that did not meet these rigorous standards ($n = 12$) due to excessive artifacts or blurring were promptly excluded. As a result, a collection of 300 top-quality CBCT scans was selected for the final analysis, highlighting our steadfast dedication to excellence.

### Inclusion and exclusion criteria

Only permanent mandibular first and second premolars were included in the CBCT images that were chosen. The inclusion criteria were teeth free of any abnormalities, including open apices, restorations, internal resorption, periapical lesions, attrition, abrasion, erosion, or any developmental anomaly. These standards were used to guarantee the integrity and homogeneity of the tooth samples during study.

### CBCT measurements

The CBCT images were assessed using the CS 3D imaging software (Carestream CS 3D Imaging, Version 3.9.3), which facilitated precise estimation of tooth dimensions. Measurements for the total tooth length, crown length, and root length were conducted in both the buccal and sagittal planes. Specifically:

- Crown length: The distance that exists between the buccal cementoenamel junction's (CEJ) a reference point and cusp tip.
- Root length: The distance between the root's apex and the point of reference at the buccal CEJ.
- Total length: The distance from the buccal cusp tip to the apex of the root.

The representation of measurements performed is depicted in Fig. 1.

### Statistical analysis

Microsoft Excel 2013 (Microsoft Corporation, Redmond, WA, USA) was used to first compile the acquired data. With the aid of IBM SPSS Version 21 (IBM Corp., Armonk,

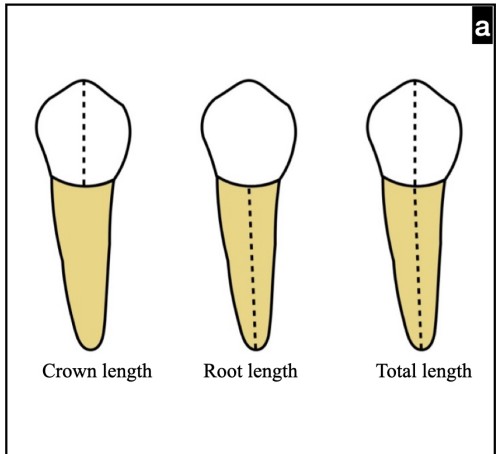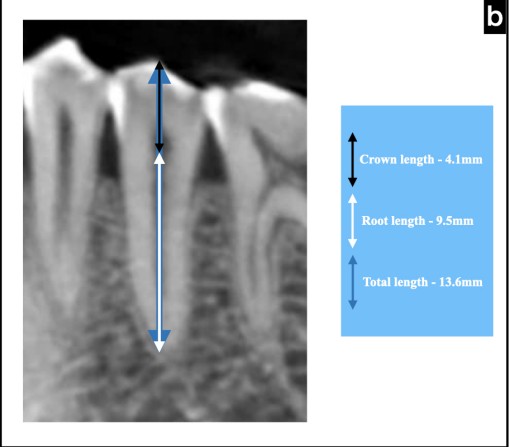

**Figure 1** Schematic representation of mandibular premolar for measuring the crown, root, and total length.

NY, USA), statistical analysis was carried out. Distributions based on frequency and percentage were computed for categorical variables like gender. Frequency and percentage distributions were calculated for categorical variables such as gender and age, followed by crown, root, and total length were continuous variables that were outlined using mean and standard deviation (SD). A one-sample $t$-test was performed to evaluate the tooth dimensions in comparison to Wheeler's standard reference values. For each statistical test, the confidence interval was set at 95%. Statistical significance was defined as a $p$-value of $<0.05$. Appropriate modifications were made to allow for the potential for multiple comparisons.

## Measurement reliability

To ensure the precision of the measurements, a senior endodontist (A.M.P.) and an oral radiologist (S.S.), independently evaluated 30 randomly selected scans for crown, root, and total length. The inter-rater reliability was determined using the Intraclass Correlation Coefficient (ICC), which indicated excellent consistency with an ICC of 0.92. For intra-rater reliability, each examiner conducted a second round of measurements on the same 30 scans after a two-week interval, resulting in an ICC of 0.95. All measurements were carried out using standardized anatomical landmarks and calibrated measuring tools incorporated into the CS 3D imaging software. Any discrepancies that arose were resolved through consensus with the assistance of a third senior endodontist (M.I.K.).

## Data handling

Through meticulous data collection and validation methods, the number of missing data was lowered. If any outliers were found, they were checked for correctness and relevance. Processes for cleaning data were used to keep it clean. The study intends to give precise odontometric data for Permanent Mandibular First and Second Premolars in the Indian Population utilizing CBCT technology through this thorough technique. The methodical

**Table 1 Comparison of crown, root, and total tooth lengths (mm) of mandibular premolars with Wheeler's dimensions.**

| Dimensions | n | Mean | Standard deviation | Wheeler's dimensions | Mean difference | t | *P* value |
|---|---|---|---|---|---|---|---|
| | | | | Mandibular first premolars | | | |
| Crown length | 300 | 6.2 | 0.78 | 8.5 | −2.3 | −48.990 | 0.000 |
| Root length | 300 | 12.9 | 1.02 | 14 | −1.1 | −17.975 | 0.000 |
| Total length | 300 | 19.1 | 1.03 | 22.5 | −3.3 | −54.850 | 0.000 |
| | | | | Mandibular second premolars | | | |
| Crown length | 300 | 6.2 | 0.80 | 8 | −1.8 | −37.623 | 0.000 |
| Root length | 300 | 13.3 | 1.39 | 14.5 | −1.2 | −14.144 | 0.000 |
| Total length | 300 | 19.5 | 1.28 | 22.5 | −3 | −38.467 | 0.000 |

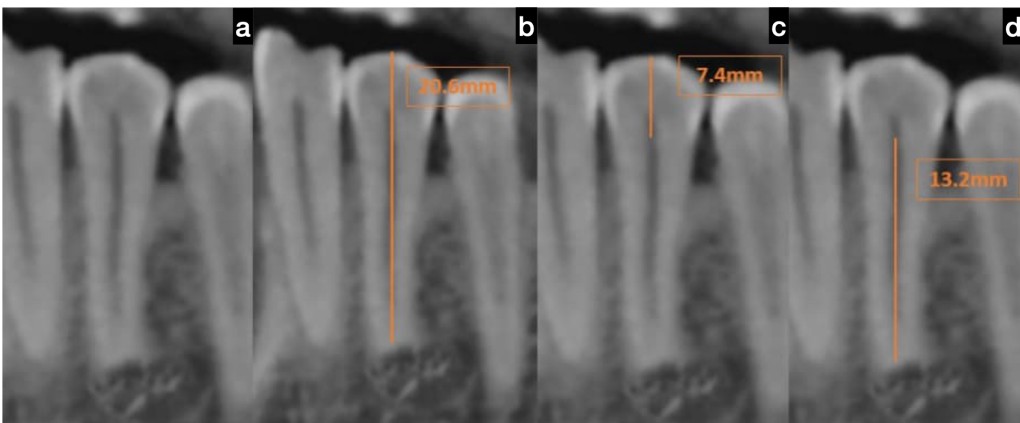

**Figure 2 Cone beam computed tomography (CBCT) scan of the mandibular first premolar.** (A) Representative sagittal CBCT image showing the orientation and location of the mandibular first premolar selected for measurement, (B) measurement of total tooth length from the buccal cusp tip to the apex of the root, indicated by a linear measuring line, (C) measurement of crown length from the buccal cementoenamel junction (CEJ) to the cusp tip, and (D) measurement of root length from the buccal CEJ to the root apex. Each measurement was conducted using standardized anatomical landmarks and calibrated CBCT imaging software.

strategy taken in gathering and analyzing the data assures the validity and applicability of the study's conclusions.

## RESULTS

This research analyzed 300 CBCT scans, consisting of 172 males and 128 females, with an average age of 41.9 ± 11.92 years, to evaluate the crown, root, and total lengths of the mandibular first and second premolars. The measurements collected were compared to Wheeler's standard dimensions using a one-sample *t*-test (see Table 1; Figs. 2, 3).

### Mandibular first premolar dimensions

There were statistically significant differences ($p = 0.000$) observed between the study group and Wheeler's values, highlighting a notable deviation that warrants attention. The average

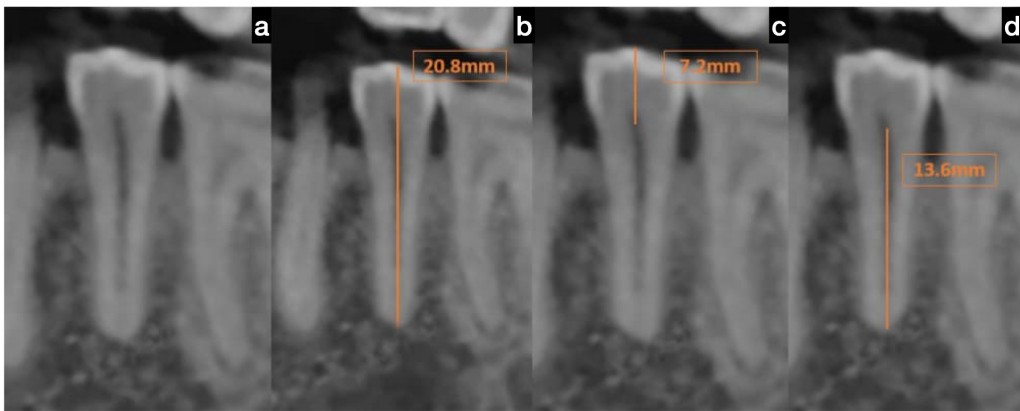

**Figure 3  Cone beam computed tomography (CBCT) scan of the mandibular second premolar.** (A) Representative sagittal CBCT image of the mandibular second premolar demonstrating the selected tooth for measurement, (B) measurement of total tooth length from the buccal cusp tip to the apex of the root, (C) measurement of crown length from the buccal CEJ to the cusp tip, and (D) measurement of root length from the CEJ to the root apex. The landmarks and measurement methodology were consistent with those used for the first premolar to ensure data reliability and reproducibility.

crown length measured just 6.2 mm, while the root length was 12.9 mm, resulting in a total tooth length of 19.1 mm. These dimensions were considerably smaller than Wheeler's measurements, with the crown being 2.3 mm shorter, the root 1.1 mm shorter, and the total length 3.4 mm shorter. This significant discrepancy calls for further exploration, as it challenges established standards and could lead to new discoveries in dental research.

## Mandibular second premolar dimensions

There were similarly statistically significant differences observed ($p = 0.000$). The average crown length measured 6.2 mm, the root length was 13.3 mm, and the overall length totalled 19.5 mm. These measurements were smaller than Wheeler's standards by 1.9 mm for the crown, 1.2 mm for the root, and 3.0 mm for the total length.

## Clinical significance

The differences highlighted in Table 2 are not just statistically significant; they hold substantial clinical importance. This persuasive evidence emphasizes the urgent need to reevaluate the dependence on conventional Western reference data, which may result in considerable errors in treatment planning for Indian patients. It is crucial that we recognize and address these variations to ensure accurate and effective medical care.

## DISCUSSION

Variations in odontometric measurements across different populations have consistently posed challenges in dental diagnostics and treatment planning. Traditional reference standards, such as Wheeler's measurements, are predominantly based on Western populations and may not accurately reflect the anatomical features of other ethnic groups. This study sought to address this issue by analyzing the crown, root, and total lengths

**Table 2 Mean difference of crown, root and total length of the root in mm with Wheeler's dimensions for mandibular first and second premolars.**

| Dimensions | Mean length | Wheeler's length | Mean difference |
|---|---|---|---|
| Mandibular first premolar | | | |
| Crown length | 6.2 | 8.5 | −2.3 |
| Root length | 12.9 | 14 | −1.1 |
| Total length | 19.1 | 22.5 | −3.4 |
| Mandibular second premolar | | | |
| Crown length | 6.2 | 8 | −1.8 |
| Root length | 13.3 | 14.5 | −1.2 |
| Total length | 19.5 | 22.5 | −3 |

of mandibular first and second premolars in an Indian population using CBCT and comparing these measurements with Wheeler's data. The null hypothesis suggested that there would be no statistically significant difference between the measured odontometric dimensions and Wheeler's reference values. However, the results led to the rejection of the null hypothesis. Statistically and clinically significant differences were found in all three measured dimensions, highlighting the need for population-specific odontometric reference values to improve diagnostic accuracy and patient outcomes.

In the Indian population, the measured lengths of crowns, roots, and overall teeth were consistently shorter than those documented in Wheeler's data. Specifically, for first premolars, the crown was 2.3 mm shorter, the root was 1.1 mm shorter, and the total length was 3.4 mm shorter. Similarly, for second premolars, the differences were 1.9 mm, 1.2 mm, and 3.0 mm, respectively. These results not only highlight population-specific variations but also have significant clinical implications. In the field of endodontics, for instance, relying on inaccurate reference values could result in misjudging working lengths, potentially leading to inadequate debridement or over-instrumentation (*Alovisi et al., 2020*).

The orthodontic implications of these variations are equally important. When dealing with cases of shorter root lengths, practitioners need to exercise caution when applying orthodontic forces, especially during intrusive and torque movements, to reduce the likelihood of root resorption. These findings highlight the necessity for orthodontic treatment plans that are tailored using precise, population-specific anatomical data *Kim et al. (2013)*. Such information is also advantageous in prosthodontics and implant dentistry, particularly for selecting the right crown sizes and determining implant lengths to ensure biomechanical compatibility and long-term success (*Magne, Gallucci & Belser, 2003*).

Beyond the confines of clinical practice, these findings carry considerable weight for the field of forensic dentistry. In scenarios such as mass disasters, cases involving mutilation, or when unidentified remains are encountered, utilizing odontometric data from populations that do not accurately represent the subject can lead to errors in identification. Establishing odontometric standards that mirror the inherent morphology of distinct ethnic groups can

significantly enhance the accuracy and trustworthiness of forensic analyses (*Kolude et al., 2010*; *Jayakrishnan, Reddy & Vinod Kumar, 2021*).

From an anthropological perspective, the noted variations illustrate the influence of evolutionary, genetic, and environmental factors specific to the Indian subcontinent. Teeth, being among the most durable and genetically consistent human tissues, are ideal for examining ancestral heritage, population movements, and environmental adaptations (*Monson, Fecker & Scherrer, 2020*). Differences in tooth size and shape have been linked to dietary practices, chewing functions, and selective pressures across generations (*Popowics & Mulimani, 2023*). Population-specific odontometric data also play a crucial role in understanding sexual dimorphism and estimating age. The dimensions of crowns and roots in mandibular and maxillary teeth, including premolars, have been effective in determining gender (*Yuwanati, Karia & Yuwanati, 2012*; *Nagare et al., 2018*). While this study does not primarily focus on this aspect, our findings encourage further exploration of sex-based odontometric differences within Indian populations.

CBCT has revolutionized dental morphometrics as an imaging technique, largely because it can produce high-resolution, three-dimensional images of dental structures. In contrast to conventional two-dimensional imaging methods, CBCT offers spatial data without distortion, allowing for precise measurements essential for both clinical and academic purposes (*Karatas & Toy, 2014*; *Venkatesh & Elluru, 2017*). This study leveraged these technological advancements by employing CS 3D Imaging software (Version 3.9.3), which facilitated the accurate and consistent measurement of tooth dimensions.

To ensure methodological rigor, we applied strict criteria for image quality control, which involved excluding scans that contained artifacts or had poor visibility of anatomical landmarks. Additionally, we confirmed inter- and intra-rater reliability with high intraclass correlation coefficients (ICC = 0.92 and 0.95, respectively), guaranteeing that the measurements were both consistent and reproducible. These steps are in line with best practices in radiographic research and bolster the credibility of the results (*Van Leeuwen et al., 2022*). Another key strength of this research lies in its unwavering commitment to methodological accuracy, which greatly enhances the study's validity and reproducibility (*Supriyono et al., 2024*). The process of selecting scans involved applying objective image quality control criteria, ensuring that only high-resolution, artifact-free CBCT images were included. Additionally, the use of standardized measurement protocols, clearly defined anatomical landmarks, and careful software versioning further support reproducibility. The study's evaluations of inter- and intra-rater reliability showed excellent agreement, highlighting the consistency and reliability of the measurements. Together, these strategies effectively reduce potential biases and enhance the robustness of the study's odontometric data.

Technological advancements continue to enhance odontometric analyses. Research has demonstrated the efficacy of digital 3D reconstructions in forensic dentistry and morphometric assessments (*Liu et al., 2016*; *Johnson et al., 2019*). These digital methodologies allow for non-destructive examination of crown and root volumes, surface areas, and dimensional ratios, thereby offering an additional level of analytical precision and facilitating comparative studies across diverse populations and regions.

Although CBCT is a valuable tool in odontometric analysis due to its non-invasive nature, three-dimensional visualization capabilities, and clinically acceptable resolution, it does have its limitations. One major drawback is its comparatively lower resolution when compared to micro-computed tomography (micro-CT). While micro-CT is limited to *in vitro* studies because of its high radiation exposure, it provides extremely detailed imaging that is perfect for analyzing microstructures (*Assari, Al Bukairi & Al Saif, 2024*). Moreover, magnetic resonance imaging (MRI), though still in the early stages of being adopted for hard tissue imaging, completely avoids ionizing radiation and is being increasingly investigated for its potential in dental diagnostics (*Al-Haj Husain et al., 2024*). However, the use of magnetic resonance imaging (MRI) for precise odontometric measurements is currently restricted due to difficulties in imaging enamel-dentin interfaces (*Idiyatullin et al., 2011*). Additionally, factors such as patient positioning, variability in voxel size, and image artifacts can introduce methodological bias in measurements based on CBCT. These factors are important to consider when interpreting CBCT data and comparing results across different populations or imaging techniques.

For precise diagnosis, treatment planning, and effective implementation in an array of dental specialties, including endodontics, prosthodontics, orthodontics, and periodontal surgery, it is essential to understand dental anatomy, morphology, and metric analysis (*Rodriguez Betancourt et al., 2023*). By showing relationships between tooth size and canal form using CBCT analysis, (*Kulkarni et al., 2020*) highlighted the importance of maxillary first premolars in this context. Understanding odontometric characteristics is essential in endodontics for procedures like root canal therapy. A complete understanding of dental anatomy and morphology is essential for successful endodontic treatment, with preoperative radiographs playing a crucial role, according to a study of maxillary and mandibular premolars in a Turkish population (*Bulut et al., 2015*).

The odontometric variations observed, notably the shorter crown, root, and overall lengths in the Indian population compared with Wheeler's measurements, have significant clinical consequences. In orthodontics, precise evaluation of crown and root lengths is crucial for biomechanical planning. Shorter roots can restrict safe tooth movement and increase the likelihood of root resorption during orthodontic procedures (*Consolaro, 2019*). This is especially relevant when applying intrusive or torque forces to the premolars, which depend significantly on root support and periodontal anchorage (*Seidel et al., 2023*). Understanding these population-specific morphometric characteristics will help orthodontists tailor treatment mechanics, anchorage strategies, and treatment duration for the individual. In endodontics, determining the root length with precision is crucial for accurately estimating the working length and for the proper shaping and filling of the canal (*Ibrahim, Ali & Mannocci, 2025*). This research points out that relying on outdated measurement standards can result in either over-instrumentation or under-instrumentation. The consistent finding of shorter root lengths underscores the need to create reference charts specific to different populations or to use CBCT-guided assessments before starting endodontic procedures. Additionally, a decrease in root length can affect the long-term success of teeth that have undergone root canal treatment, making them

more susceptible to vertical root fractures or apical issues. Recognizing these differences is vital for enhancing clinical results and reducing procedural errors.

This research highlights notable odontometric differences in mandibular premolars within the Indian demographic, highlighting the critical need for dental reference standards tailored to specific populations. The results have significant implications for refining treatment strategies in fields such as endodontics, orthodontics, prosthodontics, and forensic identification. CBCT has been pivotal as a diagnostic tool, offering high-quality, consistent morphometric data, and will remain essential in broadening our comprehension of dental anatomy across different populations. Future investigations should aim to examine other types of teeth and incorporate analyses based on regional and gender distinctions to further enrich the field of odontometric science.

While this study focused on overall odontometric means, there were 172 males and 128 females in the sample. Although this paper did not conduct a gender-based stratification in the analysis, previous literature has shown there is sexual dimorphism in crown and root dimensions (*Ajmal et al., 2023*), so future studies need to examine these dimensions to enhance clinical guidelines that are gender-specific.

It is clinically imperative that we reconsider the use of reference norms, from the West, in the contexts of treatment of patients in India. For example, with respect to endodontics, underestimating the length of the roots may lead to the practitioner failing to adequately prepare the entire canal system for obturation, while overestimating root length can increase the risk of perforated canals or separating instruments. In orthodontics, if root lengths are shorter, it requires these forces to be applied more conservatively to minimize the chance of resorbing the roots. Similarly, when planning for prosthodontics and implants, being aware of these potential variations to ensure that crown-to-root ratio and gain an optimal biomechanical result should be acknowledged. Overall, embracing the dentoskeletal morphometric characteristics of Indigenous populations and adapting treatment will facilitate more predictable, and thus safer, dental treatment.

Although this study uses CBCT to provide useful information about the odontometric dimensions of mandibular premolars in an Indian population, it must be noted that it has some of limitations. The results may not be as generalizable to larger populations due to the retrospective and single-center methodology, though. The research did not stratify data according to age or gender, two factors that are known to affect tooth morphology, despite the fact that a sizable sample size was examined. Additionally, the range of use is restricted by the exclusive focus on mandibular first and second premolars. Additional tooth kinds and the use of stratified analyses to evaluate the impact of demographic factors should be explored in future studies.

## CONCLUSIONS

In this study, CBCT scans were utilized to gather data on the crown length, root length, and total length of mandibular first and second premolars in the Indian population. A comparison with Wheeler's reference dimensions revealed consistent morphometric differences between the Indian population and Western-derived standards. These findings

carry significant implications for clinical dentistry, forensic identification, anthropological analysis, and genetic research, underscoring the importance of establishing population-specific odontometric reference values.

### Funding

This work was supported by the Deanship of Scientific Research and Graduate Studies at King Khalid University, Abha, Saudi Arabia, through the Large Research Group Project under grant number (RGP2/618/46). The funders had no role in study design, data collection and analysis, decision to publish, or preparation of the manuscript.

### Grant Disclosures

The following grant information was disclosed by the authors:
Deanship of Scientific Research and Graduate Studies at King Khalid University, Abha, Saudi Arabia, through the Large Research Group Project: RGP2/618/46.

### Competing Interests

Ajinkya M. Pawar and Luca Testeralli are Academic Editors for PeerJ.

### Author Contributions

- Fiza Hamdulay conceived and designed the experiments, performed the experiments, analyzed the data, prepared figures and/or tables, and approved the final draft.
- Ajinkya M. Pawar conceived and designed the experiments, analyzed the data, prepared figures and/or tables, and approved the final draft.
- Shivani Singh performed the experiments, analyzed the data, prepared figures and/or tables, and approved the final draft.
- Suraj Arora analyzed the data, authored or reviewed drafts of the article, and approved the final draft.
- Luca Testarelli analyzed the data, authored or reviewed drafts of the article, and approved the final draft.
- Asok Mathew analyzed the data, authored or reviewed drafts of the article, and approved the final draft.
- Alexander Maniangat Luke analyzed the data, authored or reviewed drafts of the article, and approved the final draft.
- Mohmed Isaqali Karobari analyzed the data, authored or reviewed drafts of the article, and approved the final draft.

### Human Ethics

The following information was supplied relating to ethical approvals (i.e., approving body and any reference numbers):

The Institutional Ethics Committee at Nair Hospital Dental College granted approval for the retrospective investigation (Clearance ID: EC-208/CONS/ND111/2023; Approval Date: 01-03-2023).

## Patent Disclosures

The following patent dependencies were disclosed by the authors:

The odontometric data has been registered with the Copyright Office of India (Registration of Copyright No.: L-131475/2023) to acknowledge its originality and significance, as no comparable data currently exists in the literature.

## Data Availability

The raw measurements of the samples are available in the Supplemental Files.

## Supplemental Information

Supplemental information for this article can be found online at http://dx.doi.org/10.7717/peerj.20039#supplemental-information.

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
