# Peer review of "Odontometric analysis of permanent mandibular first and second premolars in an Indian population using cone beam computed tomography"

_PeerJ, doi:10.7717/peerj.20039_

## Round 0.1 · original submission · Major Revisions

We have received four comprehensive reviews of the work.

**Language Note:** The review process has identified that the English language must be improved. PeerJ can provide language editing services - please contact us at [email protected] for pricing (be sure to provide your manuscript number and title). Alternatively, you should make your own arrangements to improve the language quality and provide details in your response letter. – PeerJ Staff

·

Basic reporting

Language and Clarity: The article demonstrates clear, professional, and technically sound language throughout. Minor grammatical improvements could be made to enhance clarity in sections like the introduction and discussion. Examples include awkward phrasing such as "considerable variances" (could be refined to "significant variations").
Context and Literature: The literature review provides relevant references and sufficient background on odontometric studies and CBCT applications in clinical and forensic contexts.
Structure: The manuscript follows the standard scientific structure—Abstract, Introduction, Methods, Results, Discussion, and Conclusion—conforming to PeerJ's guidelines.
Figures and Tables: Figures and tables are relevant and properly labeled. However, more detailed descriptions in the legends for Figures 2 and 3 would be beneficial to improve reader comprehension.
Raw Data: Raw data has been appropriately provided, aligning with PeerJ’s transparency requirements.

Suggested Improvement: Consider reviewing the language for minor grammatical revisions and enhancing figure descriptions.

Experimental design

Originality: The study presents original primary research, examining odontometric data using CBCT within the journal’s aims and scope.
Research Question: The research question is well-defined and addresses a relevant knowledge gap concerning population-specific dental morphology.
Methods: The methodology is detailed, ensuring reproducibility. Measurements for crown length, root length, and total length using CBCT scans are clearly described. Inclusion and exclusion criteria are appropriate and ensure data homogeneity.

Suggested Improvement: Include a brief discussion on the limitations of using CBCT compared to other emerging imaging techniques to address any methodological biases.

Validity of the findings

Data and Analysis: Statistical analyses, including t-tests and confidence intervals, are correctly applied to compare measurements with Wheeler’s data. Data presentation in tables is clear and interpretable.
Soundness of Conclusions: The conclusions are well-aligned with the data and highlight the clinical, anthropological, and forensic implications. The study appropriately acknowledges limitations regarding sample size and potential measurement errors.

Suggested Improvement: Provide more context on the clinical significance of the observed differences between Indian populations and Wheeler’s dimensions, particularly in orthodontics and endodontics.

Additional comments

- The study is a valuable contribution to odontometric research and its interdisciplinary applications. The discussion section effectively bridges clinical applications and evolutionary implications, but a more concise summary could improve readability.
- Ethical considerations are clearly addressed, with approval from the relevant ethics committee and anonymization of data.

Reviewer 2 ·

Basic reporting

Abstract
Conciseness and clarity are required, and repetitions should be avoided. For example removes redundancy and uses more precise language. Phrases like "the intent of this investigation was to obtain" are replaced with more direct statements.
Introduction
This text is a good introduction to a research paper on odontometry. However, it could benefit from some restructuring and tightening. Here are some suggestions for improvement:

Overall Structure: The introduction currently feels somewhat disjointed. It jumps between different applications of odontometry without a clear, logical flow. A more effective structure would establish a clear problem statement (the limitations of traditional methods) and then present the proposed solution (using CBCT) and its significance.
Combine Sections: The paragraphs discussing the importance of odontometry in different fields (forensics, anthropology, education, clinical practice) could be merged and organized thematically. For instance, you could group together the applications in clinical dentistry, then those in forensic science, and finally those in anthropological research.

Redundancy: Several points are repeated across different sections. For example, the applications of odontometry in various dental specialties are mentioned multiple times. Consolidating this information would improve clarity and conciseness. The benefits of CBCT are mentioned multiple times. Summarize these benefits in one or two sentences early in the introduction, then expand on specific applications later as needed.
Explicitly state the research gap. What is currently unknown or poorly understood about mandibular premolar odontometry, particularly in the Indian population? This should be stated clearly before presenting your solution.

Experimental design

Materials and methods are well explained and supported by figures. But;
CBCT Image Quality Control: The statement "300 CBCT scans, all of which were found to be of excellent quality for analysis" is subjective. What specific criteria were used to define "excellent quality"? Were any images excluded due to poor quality, and if so, how many and why? Objective quality control measures should be included.
Measurement Error: The methods describe the measurements but do not address potential sources of error (inter-rater reliability, intra-rater reliability, measurement precision). Did multiple examiners perform measurements? If so, was inter-rater reliability assessed? Were measurements repeated by the same examiner to assess intra-rater reliability? Addressing these potential sources of error is essential for ensuring the reliability of the results.
Software Details: While the software used (CS 3D) is mentioned, the version is not specified. This seemingly minor detail is important for reproducibility.
In summary, while the revised Materials and Methods section is an improvement, enhancing these areas would substantially increase the rigor and reproducibility of the study. Adding detail and addressing potential biases and sources of error is crucial for strengthening the credibility of the research.

Validity of the findings

In the result sections, I suggest:
This study analyzed 300 CBCT scans (172 male, 128 female; mean age 41.9 ± 11.92 years) to measure crown length, root length, and total length of mandibular first and second premolars. Measurements were compared to Wheeler's established dimensions (Tables 2 and 3; Figures 2 and 3).

Mandibular First Premolars: Statistically significant differences (p = 0.000) were observed between the study population and Wheeler's dimensions. Mean values in the study population were: crown length 6.2 mm, root length 12.9 mm, and total length 19.1 mm. These values were consistently smaller than Wheeler's dimensions by 2.3 mm (crown), 1.1 mm (root), and 3.4 mm (total length), respectively.

Mandibular Second Premolars: Similar statistically significant differences (p = 0.000) were found for mandibular second premolars. Mean values in the study population were: crown length 6.2 mm, root length 13.3 mm, and total length 19.5 mm. These were smaller than Wheeler's dimensions by 1.9 mm (crown), 1.2 mm (root), and 3.0 mm (total length), respectively.

Clinical Significance: The observed differences in crown, root, and total lengths between the Indian population and Wheeler's reference values (summarized in Table 4) were statistically and clinically significant.
Note:
The revised results section is more effective in communicating the key findings of the study in a clear and concise manner. It highlights the main differences between the study group and the reference data and emphasizes their statistical and clinical significance. Table 1 is not required. Ensure Tables 2, 3, and 4, as well as Figures 2 and 3, are correctly referenced and included in the manuscript. Consider combining Tables 2 and 3 into a single table for improved efficiency.

·

Basic reporting

Global comments
This work addresses a pertinent clinical issue.
Make a full review of the writing, it must necessarily be improved.
Avoid repetition of concepts.

Comment 1 – Title
None.
Comment 2 - Abstract
"Considering that the abstract is the section of the article designed to convince the reader to engage with the entire content, write the abstract in a way that piques curiosity." Therefore:
a) In the Background, write at least one phrase to introduce the problem to the reader before stating the aim of the study.
b) Specify the specific data analysis performed instead of making general considerations.
c) Keep writing in a technical way. Writing should be simpler and more focused.
d) Use p < 0.001 instead of p=0.000 or describe the exact value, such as p = 0.0001, if the software you use provides this precision. Zero probability of error is statistically not possible.
e) Rephrasing suggestion: "The odontometric findings of this study highlight the unique dental morphology of the Indian population, offering valuable insights with significant implications for anthropology, evolutionary biology, forensic science, and clinical dentistry."

Comment 3 – Introduction
a) In line 54, what do you mean by “operations”?
b) From lines 59-65, I suggest merging sentences to avoid repetition of concepts.
c) In line 69, I suggest changing “instills” to “implies”. I think it is more understandable to everybody.
d) Lines 108-110, please review and use an impersonal way of writing.
e) In line 110, I suggest changing “comprehending” to ”Understanding”
f) Try merging lines 112 to 136.
g) Why was there no formulation of a null hypothesis?

Experimental design

Comment 4 - Materials and Methods
a) Cut “From March 2023 until October 2023, this study was executed out,” in line 139, as it will be stated below.
b) Rephrase by merging: “As part of standard radiographic evaluations for various dental procedures, these CBCT images were gathered. The scans were obtained utilizing a CBCT scanner (NewTom VGi, Verona, Italy), as it is obvious that a CBCT is obtained by a CBCT scanner.
c) Please explain which were the selection criteria to select the CBCT; were they randomly selected? Was the population, anyhow, stratified? Were the patients adults? Male? Females? Please specify in this section the criteria to ensure the characteristics of the sample.
d) Rephrasing suggestion:
“Using CS 3D imaging software, which enabled precise estimation of tooth dimensions, CBCT images were analyzed. Measurements of total tooth length, crown length, and root length were taken in both the buccal and sagittal planes as follows:
Crown Length: The distance from the reference point at the buccal cementoenamel junction (CEJ) to the cusp tip.
Root Length: The distance from the root apex to the reference point at the buccal CEJ.
Total Length: The distance from the buccal cusp tip to the root apex.
Figure 1 illustrates the methods used for these measurements.
e) Please complete the brand and place of production for Excel and SPSS.
f) Consider. “Frequency and percentage distributions were calculated for categorical variables such as gender.”
g) Consider: "A one-sample t-test was conducted to compare tooth dimensions against Wheeler's categorization.”

Comment 5 – Results
a) To be accurate, it should be “300 CBCTs obtained from……” with an average age of 41.9±11.92 years.
b) Consider: When comparing the crown, root, and total lengths of the mandibular first premolar (Figure 2) to Wheeler's dimensions, significant differences (p <0.001) in mean measurements were observed among the Indian population.
c) Remove the next sentence.
d) Apply the same philosophy in the section Mandibular Second PM
Comment 6 – Discussion

Validity of the findings

LINES 226-240 DO NOT BELONG TO THE CONTEXT OF THE ARTICLE.
Begin this section by reminding the reader about the problem to be studied. Then remind the aim of the study, and refer to the null hypothesis, its rejection or acceptance.
Restructure the entire discussion, as there is a lot of repetition of statements, which ends up tiring the reader. Focus on pertinent ideas.
- In lines 320, 322, the citation must be corrected.
Comment 7 – Conclusions
Consider replacement by: “In this study, cone beam computed tomography (CBCT) scans were utilized to gather data on the crown length, root length, and overall length of mandibular first and second premolars in the Indian population. A comparison of this data with Wheeler's standard dimensions highlighted morphometric differences between the Indian population and an unspecified population. These findings have significant implications for various fields, including clinical dentistry, anthropology, genetics, and forensics.”

Additional comments

Comment 6 – References
No comments

FIGURES
If possible, provide higher resolution.

·

Basic reporting

1. This research is too simple to be published:
A. In this study, only the dimensions of the first and second premolar teeth were compared. Why not measure all types of teeth, because each type of tooth is very specific and has the same role as the first and second mandibular premolar teeth?
B. Tooth dimensions are influenced by several factors, such as age, gender. This study should also examine the relationship between the dimensions of the teeth of the Indian population with gender and age factors.

Experimental design

2. The Introduction, in general, is too long,
for example:
A. Explanation of Odontometry (lines 51-73; 125-130), the author should write it more concisely in one paragraph.
B. Explanation of the importance of mandibular premolar teeth is written in lines 73-80, but the same thing is written again in lines 105-111.
C. Explanation of CBCT is too long (lines 81-104); the author should make it shorter in one paragraph.
D. On lines 112-136, the author should only write about the purpose of this study

Validity of the findings

3. In the Discussion section, there are statements that are not appropriate, for example:
Lines 226-240: Is this explanation appropriate and related to the research being conducted?

4. The statement on lines 327-341 requires references

5. Table 4 is not needed because all the data in Table 4 is already in Tables 2 and 3

---

## Round 0.2 · Minor Revisions

·

Basic reporting

NA

Experimental design

NA

Validity of the findings

NA

Additional comments

NA

·

Basic reporting

The manuscript is generally well-written and employs professional English throughout.
The introduction offers a strong interdisciplinary context, covering dental, forensic, and anthropological relevance.
Figures and tables are clearly labeled and described, with appropriate legends.
Raw data appears to be supplied and is consistent with the results presented.

Experimental design

The design is methodologically sound and fits the scope of PeerJ.
Clear inclusion and exclusion criteria were defined.
Sample size calculation was performed and justified based on previous work.
CBCT protocols, imaging quality control, and measurement standardization are exemplary.

Validity of the findings

Statistical methods are appropriate (e.g., one-sample t-test), and the differences are robust (p < 0.001 throughout).
Intra and inter-rater reliability is excellent (ICC = 0.92–0.95).
Data presentation is clear and supported by meaningful visuals.

Additional comments

This article is a well-executed and technically solid study that offers valuable, population-specific odontometric data using a modern imaging technique. The manuscript meets the standards of originality, scientific rigor, and clarity.
However, the manuscript would be substantially improved by condensing and refining the introduction, including at least a brief gender-based analysis, expanding on the clinical recommendations based on the findings (i.e., how dentists should modify treatment planning for Indian patients).

·

Basic reporting

1. In the Introduction, Paragraph 1-4 ( line 52-86) that are talking about odontometry should be shortened to 2 paragraph.
2. In the Introduction, Paragraph 5-8 (line 87-124) that are talking about CBCT should also be shortened
3. These sentences:
Among the various teeth, mandibular premolars, particularly the first and second, are of considerable clinical significance due to their roles in chewing, maintaining occlusal balance, and serving as anchorage in orthodontic procedures (Albuquerque,Kottoor & Hammo, 2014). The morphological variations of these teeth influence the design of dental restorations and the maintenance of occlusal harmony, thereby affecting overall dental functionality (Sierpinska et al., 2017).

should be combined with paragraph 9 (line 125-135).

4. In the Introduction, Paragraph 9-10 (line 136-153) should be shortened, just focus on what is the aim/purpose of the study.

Experimental design

Research question well defined, relevant & meaningful, and Methods described with sufficient detail & information to replicate.

Validity of the findings

All underlying data have been provided; they are robust, statistically sound, & controlled.

Conclusions are well stated, linked to original research question & limited to supporting results.

Additional comments

-

---

## Round 0.3 · Minor Revisions

Reviewer 4 suggests some minor revisions.

·

Basic reporting

No comment

Experimental design

No comment

Validity of the findings

The authors made the requested changes.

Additional comments

No comments

·

Basic reporting

1. In the Introduction, Paragraph 1-4 ( line 52-86) that are talking about odontometry should be shortened to 2 paragraph.
2. In the Introduction, Paragraph 5-8 (line 87-124) that are talking about CBCT should also be shortened.
3. These sentences:
Among the various teeth, mandibular premolars, particularly the first and second, are of considerable clinical significance due to their roles in chewing, maintaining occlusal balance, and serving as anchorage in orthodontic procedures (Albuquerque,Kottoor & Hammo, 2014). The morphological variations of these teeth influence the design of dental restorations and the maintenance of occlusal harmony, thereby affecting overall dental functionality (Sierpinska et al., 2017).

should be combined with paragraph 9 (line 125-135).

4. In the Introduction, Paragraph 9-10 (line 136-153) should be shortened, just focus on what is the aim/purpose of the study.

Experimental design

-

Validity of the findings

-

Additional comments

-

---

## Round 0.4 · accepted · Accept

Dear Dr. Pawar, I am pleased to inform you that this article has been accepted for publication.

·

Basic reporting

The introduction was clear and unambiguous.

Experimental design

Methods are well defined, clearly described with sufficient detail and information.

Validity of the findings

All underlying data have been provided. The conclusion is also well stated.